# The Role of Lonp1 on Mitochondrial Functions during Cardiovascular and Muscular Diseases

**DOI:** 10.3390/antiox12030598

**Published:** 2023-02-28

**Authors:** Giada Zanini, Valentina Selleri, Mara Malerba, Kateryna Solodka, Giorgia Sinigaglia, Milena Nasi, Anna Vittoria Mattioli, Marcello Pinti

**Affiliations:** 1Department of Life Sciences, University of Modena and Reggio Emilia, 41125 Modena, Italy; 2Istituto Nazionale per le Ricerche Cardiovascolari, 40126 Bologna, Italy; 3Surgical, Medical and Dental Department of Morphological Sciences Related to Transplant, Oncology and Regenerative Medicine, University of Modena and Reggio Emilia, 41125 Modena, Italy; 4Department of Medical and Surgical Sciences for Children and Adults, University of Modena and Reggio Emilia, 41124 Modena, Italy

**Keywords:** Lon protease, UPRmt, heart dysfunction, sarcopenia

## Abstract

The mitochondrial protease Lonp1 is a multifunctional enzyme that regulates crucial mitochondrial functions, including the degradation of oxidized proteins, folding of imported proteins and maintenance the correct number of copies of mitochondrial DNA. A series of recent studies has put Lonp1 at the center of the stage in the homeostasis of cardiomyocytes and muscle skeletal cells. During heart development, Lonp1 allows the metabolic shift from anaerobic glycolysis to mitochondrial oxidative phosphorylation. Knock out of Lonp1 arrests heart development and determines cardiomyocyte apoptosis. In adults, Lonp1 acts as a cardioprotective protein, as its upregulation mitigates cardiac injury by preventing the oxidative damage of proteins and lipids, and by preserving mitochondrial redox balance. In skeletal muscle, Lonp1 is crucial for cell development, as it mediates the activation of PINK1/Parkin pathway needed for proper myoblast differentiation. Skeletal muscle-specific ablation of Lonp1 in mice causes reduced muscle fiber size and strength due to the accumulation of mitochondrial-retained protein in muscle. Lonp1 expression and activity decline with age in different tissues, including skeletal muscle, and are associated with a functional decline and structural impairment of muscle fibers. Aerobic exercise increases unfolded protein response markers including Lonp1 in the skeletal muscle of aged animals and is associated with muscle functional recovery. Finally, mutations of Lonp1 cause a syndrome named CODAS (Cerebral, Ocular, Dental, Auricular, and Skeletal anomalies) characterized by the impaired development of multiple organs and tissues, including myocytes. CODAS patients show hypotonia and ptosis, indicative of skeletal muscle reduced performance. Overall, this body of observations points Lonp1 as a crucial regulator of mitochondrial functions in the heart and in skeletal muscle.

## 1. Introduction

The primary function of mitochondria is to generate large quantities of ATP, but they are also involved in processes like apoptosis, autophagy, reactive oxygen species (ROS) production, or calcium handling [1,2,3]. Dysfunctional mitochondria generating less ATP have been observed in various aged organs, including skeletal muscle and heart [4]. Dysfunctional cardiac mitochondria contribute to the onset of different cardiovascular diseases, such as ischemia/reperfusion (I/R) injury, ventricular hypertrophy, cardiomyopathies, and heart failure (HF) [5,6,7]. Similarly, diverse mitochondrial changes in skeletal muscle can contribute to an age-related loss in skeletal muscle mass and a decline in skeletal muscle function, a condition defined as sarcopenia [8].

In recent years, the nuclear-encoded mitochondrial protease Lonp1 has been established as a crucial regulator of a variety of mitochondrial processes and has been identified as pivotal in the preservation of organelle balance, as well as in the response to physiological and pathological stressors [9]. As cardiomyocytes and myocytes are cells with high metabolic activity, a high number of mitochondria and a continuous production of ROS, it is not surprising that the role of Lonp1 in the homeostasis of the heart and of the skeletal muscle has been widely investigated and unveiled a crucial role of Lonp1 in the maintenance of cell homeostasis in these organs. 

In this review, we explore primarily the physiological and pathophysiological functions of Lonp1 in humans. Then, we discuss its role in heart and skeletal muscle in health and disease, emphasizing its involvement in the response to oxidative stress. Finally, we discuss the role of Lonp1 in the aging process, and the functional outcomes of the age-related Lonp1 expression decline in the onset of cardiovascular diseases and skeletal muscle functional decline. 

## 2. General Features and Functions of Lonp1 and Its Implication in Diseases

Lonp1 is a mitochondrial ATP-dependent protease that is encoded by nuclear DNA, and it is highly conserved across evolutionary lineages [10,11,12]. In humans, the LONP1 gene, which encodes this protein, is located on chromosome 19. Lonp1 is synthesized in the cytosol as a 106 kDa precursor protein, which is targeted to the mitochondrial outer membrane by an amino-terminal mitochondrial targeting sequence (MTS). After cleavage of the MTS in the matrix, the mature Lonp1 protein is mainly located in the mitochondrial matrix, although it has been recently found to be present also in the nucleus, cytosol, and endoplasmic reticulum [13,14]. The active form of Lonp1 consists of three distinct domains: the N domain for substrate recognition and binding, a conserved AAA+ domain for ATP binding and hydrolysis, and the highly conserved P domain containing the serine–lysine dyad that forms the enzyme proteolytic active site [15]. Lonp1 is active in the mitochondrion as a hexameric complex [16]. Although Lonp1 is ubiquitously expressed, it is most highly expressed in high-energy-demanding tissues, such as the heart, skeletal muscle, brain, and liver [10]. Genetic evidence obtained from studies conducted in the yeast *Saccharomices cerevisiae*, in *Drosophila melanogaster,* and particularly in mouse models have demonstrated that the absence of Lonp1 is incompatible with life [17].

Lonp1 exerts multiple functions in the mitochondria (Summarized in Table 1). Lonp1 plays a crucial role in maintaining mitochondrial homeostasis and is essential for survival and protein quality control in vivo and in vitro [18,19]. Lonp1 principal substrates are components of the tricarboxylic acids (TCA) cycle, oxidative phosphorylation (OXPHOS) amino acid (e.g., serine hydroxymethyltransferase-2) and lipid metabolism [20]. Several lines of evidence indicate that Lonp1 work as an antioxidant for the oxidized proteins degradation in the mitochondrial matrix, and to restrict the oxidative damage under hypoxic conditions [9,21,22]. Lonp1 selectively degrades oxidized and damaged proteins, including aconitase, an essential mitochondrial enzyme, which is susceptible to oxidative damage and undergoes modification and inactivation during ageing [23]. Accordingly, in vitro studies have shown its capability, when upregulated, in maintaining normal cell viability in response to acute stress stimuli [24,25] while another study reported that the upregulation of Lonp1 causes cell death [26].

In addition, Lonp1 works with the mitochondrial heat shock protein 70 (mtHSP70) chaperone system to promote protein folding in mitochondria, and its inhibition results in protein aggregation, suggesting Lonp1 chaperone activity [27]. Matsushima et al. have shown the physical interaction between Lonp1 and the proteins of mitochondrial import machinery, mtHSP70, TIMM44, TIMM23, and TOMM40, highlighting that Lonp1 supports incoming proteins folding through its chaperone-like function and degrades abnormal imported proteins through the combination of its protease and unfolding activities. Thus, Lonp1 contributes to the solubilization of mitochondrial proteins [28]. Lonp1 overexpression has been observed in several cancers and has been shown to promote cell proliferation, apoptotic resistance to stresses, and the transformation of tumor cells [25,29,30]. 

Preservation of mitochondrial integrity is essential for the optimal functioning of cells. In order to maintain mitochondrial homeostasis, Lonp1 is also involved in the regulation of mitochondrial dynamics and mitophagy, a specific, selective form of autophagy that identifies and eliminates dysfunctional mitochondria [16]. The process of mitophagy is crucial to the breakdown of non-functioning mitochondria [31]. Depolarization of the mitochondrial membrane potential induces the accumulation of PTEN-induced putative kinase 1 (PINK1) on the outer mitochondrial membrane, thus initiating a mitophagic process that is mediated by the recruitment of the parkin RBR E3 ubiquitin protein ligase (PARK2) and other molecules, and targets mitochondria for degradation within lysosomes [32]. Different studies demonstrated that Lonp1 is implicated in the proteolytic system responsible for PINK1 degradation [33,34,35]. Mitophagy is critical for maintaining cardiovascular and muscular health and homeostasis, in agreement with the elevated energy requirements of cardiomyocytes and myocytes [7]. Recent research suggests that dysfunctions in mitochondrial dynamics and mitophagy may be implicated in the development of sarcopenia [36,37].

Consistent with its functions, the expression of Lonp1 is induced by several stress stimuli, such as hypoxia, heat shock, ER and oxidative stress, ischemic preconditioning, nutrient depletion, and unfolded protein response (UPR) [24,29,38,39]. A study by our research group highlighted Lonp1 presence in the nucleus of colorectal cancer cells undergoing heat shock and its association with enhanced expression of stress response genes, suggesting that Lonp1 is involved in cellular stress response [14]. Another work has shown Lonp1 presence at mitochondria-associated membranes (MAMs) in the ER-mitochondrial interface to preserve mitochondrial homeostasis against ER stress [13]. Moreover, during hypoxia, Lonp1 gene expression is upregulated by hypoxia inducible factor (HIF)-1α through direct binding to the Lonp1 promoter, clearly indicating that Lonp1 is required to cope with low oxygen availability [24].

In addition to its functions as chaperon and protease, Lonp1 is involved in the regulation of mitochondrial DNA copy number through direct and indirect mechanisms. Lonp1 is able to bind single-stranded DNA (ssDNA) [40,41], a property conserved in the bacteria homologue of Lonp1, named Lon, in the yeast protein homologue Pim1 and in mammals [40,41,42]. Human Lonp1 can bind mitochondrial DNA (mtDNA) thanks to the presence of G-quadruplexes located on mtDNA heavy-strand [40]. Additionally, Lonp1 selectively degrades mitochondrial transcription factor A (TFAM), a central regulator of mtDNA replication and transcription, only when TFAM is DNA-free [18]. Lonp1 can also bind single-stranded RNA sequences [43]. 

The essential role of Lonp1 in mitochondrial homeostasis has been inferred not only from studies in vitro, but also from observations made in patients bearing mutations of Lonp1. Pathogenic Lonp1 mutations are associated with CODAS syndrome, a rare multisystem disorder affecting the development of the cerebral, ocular, dental, auricular, and skeletal systems [44,45,46,47]. Clinical hallmarks of CODAS syndrome include radiographic irregularities and congenital bilateral cataracts. Notably, some affected individuals present incomplete cardiac septation and atrioventricular defects [45] and others present hypotonia and ptosis, highlighting the importance of Lonp1 in the normal functionality of muscle cells. However, since all CODAS patients are young children, the impact of these mutations on age- or stress-related disorders, such as cardiovascular diseases, has not been extensively examined.

Another disease triggered by Lonp1 mutation is PDH deficiency, characterized by profound neurodegeneration accompanied by progressive cerebellar atrophy due to cerebral lactic acidosis [48]. Skin fibroblasts from patients exhibited an increased lactate:pyruvate ratio and defects in glucose oxidative metabolism. These defects were linked to the inability of mutated Lonp1 to degrade subunit a of the E1 pyruvate decarboxylase (E1a) enzyme, which is part of the PDH complex and involved in the decarboxylation of pyruvate [48].

Recently, a different set of mutations of the Lonp1 human gene has been shown to cause congenital diaphragmatic hernia (CDH), whereby abnormal development of the diaphragm before birth leads to defects ranging from a weakened area in the diaphragm to its full absence [49].

Lonp1 expression and activity is also related to exercise and aging. While Lonp1 expression declines during aging [21], mice that undergo lifelong exercise exhibit upregulated Lonp1 expression in the heart and longer lifespan compared to sedentary ones [50]. 

Taken together, these observations depict a scenario where Lonp1 is a crucial player in the mitochondrial response to stress, particularly oxidative stress.

**Table 1 antioxidants-12-00598-t001:** Main physiological functions of Lonp1 in the mitochondria.

Lonp1 Functions	References
Selectively degrades oxidized and damaged proteins, including aconitase	[23]
Maintains normal cell viability in response to acute stress stimuli when upregulated	[24,25]
Works with mitochondrial HSP70 chaperone system to promote protein folding in mitochondria	[27]
Supports incoming proteins folding through its chaperone-like function and degrades abnormal imported proteins	[28]
Contributes to the solubilization of mitochondrial proteins	[28]
Promotes cell proliferation, apoptotic resistance to stresses	[25,29,30]
Involved in the regulation of mitochondrial dynamics and mitophagy	[16]
Implicated in the proteolytic system responsible for PINK1 degradation	[33,34,35]
Can bind mitochondrial DNA	[18]
Involved in the regulation of mitochondrial DNA copy number, through direct and indirect mechanisms	[18,40,41,42,43]

## 3. The Role of Lonp1 in Heart Homeostasis and Response to Stress

Mitochondria play a fundamental role in sustaining the cardiac contractility, and mitochondrial impairment has been implicated in several cardiovascular (CVD) diseases, including HF, by causing a decrease in ATP production and thus a cardiac energy deficiency [51]. Accordingly, risk factors of CVD diseases, such as atherosclerosis, metabolic syndrome, and diabetic hyperglycemia [52,53,54,55,56], have been correlated to mitochondrial dysfunction [57,58]. Cardiac cell apoptosis, induced by an excess of ROS under ischemia or I/R or ischemic preconditioning (IPC), is a noteworthy pathological event in the context of HF [59].

Mitochondria are the primary source of ROS, which at physiological amounts act as “redox messengers”. Yet, when abundant, they induce oxidative stress, resulting in cell death [60]. Thanks to its pivotal role in preserving mitochondrial homeostasis and in the regulation of ROS production, Lonp1 has become of great interest in CVD studies. Although there is consensus in the literature that Lonp1 expression is upregulated during injury or cardiac stress [61], it is not completely clear whether Lonp1 promotes or ameliorates cardiac injury.

Lonp1 plays a fundamental role in heart development during embryogenesis. The heart is first organ to form and become functional [62]. Initially, prior to embryonic day (E) 11.5, the heart tissues form in a hypoxic environment, and HIF1α promotes anaerobic glycolysis to provide energy for early cardiac development [63,64]. Subsequently, metabolic alterations occur in cardiomyocytes, wherein mitochondrial OXPHOS is essential for myocardial growth [64,65]. Throughout this process, Lonp1 safeguards the expression and integrity of mtDNA [9,25], prevents protein aggregation [28], monitors mitochondrial proteostasis by degrading dysfunctional and oxidized proteins, and regulates glucose oxidation [48]. Lonp1 is also necessary for regulating pyruvate dehydrogenase (PDH) activity, which is an important enzyme at the glucose and fatty acid oxidation (FAO) junction.

Genetic studies in mice have demonstrated the crucial importance of Lonp1 in cardiac development. As stated before, deletion of the Lonp1 gene in mice is lethal during early gastrulation, with mtDNA depletion invariably associated with this phenomenon [25,66]. Tissue-specific knock out of Lonp1 in embryonic cardiac tissues showed that Lonp1 loss results in severe defective heart development and embryonic lethality. Mechanistically, protein aggregates, including activating transcription factor 4 (ATF4), a master regulator of the integrated stress response (ISR), were present in the mitochondria of Lonp1-deficient cardiomyocytes [67]. This led to the expression of glycolysis regulatory genes and the subsequent disruption of the metabolic shift needed for cardiac development. 

Lonp1 is a crucial protein for energy metabolism in maturing cardiomyocytes, as it helps to regulate glucose and FAO levels. By modulating the activity of pyruvate dehydrogenase (PDH) and FAO enzymes, Lonp1 ensures that the energy demand of the cardiomyocytes in a timely manner [48]. Primary fibroblasts expressing mutated Lonp1 revealed blocked PDH activity [48], which has been shown to increase aerobic glycolysis and favor FAO [68]. 

Studies in mouse models have also shown the involvement of Lonp1 in atherosclerosis [69], and its oxidative inactivation after transaortic constriction (TAC) in mouse heart [70]. Atherosclerosis is a chronic inflammatory and metabolic disease and a major risk factor for myocardial infarction [71,72]. During atherosclerosis, Lonp1 is upregulated in macrophages by saturated fatty acids and by PERK-eIF2α activated signaling. This induces ISR and ATF4, leading to the reduction of PINK1, Parkin, and mitophagy while increasing mitochondrial ROS, inflammation, and IL-1β secretion. In fact, chronic ER stress caused by dietary fats and the activation of PERK-eIF2a-Lonp1 signaling leads to high mtROS levels, which in turn activate the inflammasome, resulting in a high inflammatory cytokine production during atherogenesis [69]. PERK inhibition decreases atherosclerotic lesions in the aorta and reduces eIF2α phosphorylation in macrophage-rich areas of plaque [69]. Moreover, PERK inhibition reduces ATF4 and Lonp1 mRNA expression in atherosclerotic plaque. In this scenario, Lonp1 upregulation was shown to exacerbate the formation of atherosclerotic plaques and promote myocardial ischemic damage. 

Interestingly, there are studies showing that Lonp1 itself is susceptible to oxidative modifications [70,73]. In vivo experiments in mice undergoing TAC demonstrated that Lonp1 activity is reduced because of oxidative post-translational modifications [70]. In addition, cardiac pressure overload induced by TAC was shown to promote Lonp1 carbonylation, cysteine reduction, and tyrosine nitrosylation, which is associated with a mitochondrial ATP-dependent proteolytic activity reduction [70]. This evidence suggests that Lonp1 plays a pivotal role in the development and progression of HF and points to endogenous Lonp1 as a potential therapeutic target for HF.

Lonp1 can also promote apoptosis in cardiomyocytes through the accumulation of ROS [26]. Studies on the effect of Lonp1 on cardiomyocyte survival under normoxic and hypoxic conditions showed that Lonp1 overexpression caused apoptosis under normoxic conditions; nevertheless, Lonp1 downregulation mitigated cell death induced by hypoxia. Under hypoxic conditions, Lonp1 is upregulated in rat cardiomyocyte H9c2 cells, a well-established cell model for studying cardiac disease in response to oxidative stress [74,75], with an increased ROS-induced apoptosis. Apoptotic signals induced by oxidative stress can result in many pathological conditions, including heart diseases, which are a consequence of ROS increment or antioxidants decrease and cause an impairment in the intracellular redox homeostasis [76,77]. Interestingly, this observation appears to be in opposition to what is seen in other cell types and models, particularly in tumour cells. For example, Lonp1 overexpression in cancer cells, such as colorectal cancer [14,21,30], Oral Squamous Cell Carcinoma, and Non-Small Lung Cancer, promotes cell proliferation, apoptotic resistance to stress, and neoplastic transformation [29]. This apparent contradiction between cancer cells and cardiomyocytes may be due to their different tolerance to ROS levels. Kuo et al. found that short hypoxia did not induce apoptosis, but it was able to activate autophagy that protects cells against death [26]. Moreover, Takahashi et al. showed that short hypoxia exposure favoured H9c2 cell growth while longer exposure caused cell apoptosis [78]. Autophagy has been shown to protect myocardium and cardiac cells against I/R injury following IPC [79,80]. Thus, Lonp1 acts as a stress protein during prolonged hypoxia exposure, causing cell death via apoptosis; however, it also acts as a “stress-sensor” protein during short hypoxia exposure and protects cells against death via autophagy. These data suggest that mitochondria can induce nonlethal oxidative stress providing an adaptive and beneficial response to promote metabolic health and longevity [81,82]. 

In contrast to what was observed in rat cell model H9c2, Lonp1 seems to have a protective role in I/R in mouse hearts. A series of insights into the role of Lonp1 in I/R came from in vivo studies with IPC. IPC is a cardioprotective experimental protocol mitigating reperfusion-induced damage of the myocardium [83]. IPC is carried out through cycles of ischemia (3 min) and reperfusion (3 min) before I/R injury surgery. Venkatesh et al. showed that Lonp1 is involved in the pathway of IPC since the cardioprotective effect of IPC in heterozygous knockout mice Lonp1^+/−^ was significantly reduced. Our group measured the expression of Lonp1 in a similar Lonp1^+/−^ mouse model, and confirmed a heart-specific marked reduction of Lonp1 in adult, but not young heterozygous, mice [66], suggesting the cardioprotective effect of Lonp1 shown by Venkatesh could vane with age. Furthermore, myocardial infarct size post I/R injury was significantly larger in Lonp1^+/−^, and significantly smaller in transgenic mice overexpressing Lonp1 [84]. IPC is mediated by HIF-1α [85], a transcriptional activator of Lonp1 [24,86], and the heterozygous knockout of HIF-1α in mice abolished IPC cardioprotective effects [85]. Thus, IPC may generate moderate ROS levels, activating a potential ROS response element within *Lonp1* promoter [87]. Defective Lonp1 increases infarct sizes after I/R injury while Lonp1, upregulated in mouse hearts during IPC, when overexpressed, can significantly reduce cardiac infarction and cell apoptosis in mice cardiomyocytes [61]. In mice, Lonp1 upregulation reduces oxidative damage caused by lipids and proteins, preserves redox state of mitochondria, and reprograms mitochondrial bioenergetics through the reduction of complex I activity, all of which leads to less ROS production and cardiac cell death [61]. In addition, a recent study highlighted that heart-specific Lonp1 deficiency in mice causes the fragmentation of mitochondria, aberrant metabolic reprogramming of cardiomyocytes, dilated cardiomyopathy, and HF [67], suggesting an essential role of Lonp1 in regulating mitochondrial dynamics and its importance in normal cardiac physiology. Thus, these lines of evidence place Lonp1 as one of the mediators of IPC cardioprotection by attenuating ROS- and ischemia-induced damage and reducing infarct size upon I/R injury. 

Lonp1 specific upregulation in cardiomyocytes and the heart has been observed in vitro and in vivo in response to cellular and cardiac perturbations [61,88]. The expression and activity of Lonp1 were also studied in a mouse model of Friedreich ataxia (FRDA), a rare disease characterized by progressive cardiomyopathy and ataxia with a deficiency in mitochondrial frataxin, which is essential for Fe-S clusters assembly. In this model, Lonp1 and caseinolytic protease proteolytic subunit (ClpP), another mitochondrial matrix protease involved in protein quality control, resulted in upregulation at mid-stage cardiomyopathy if compared to control mice [89], with a concomitant reduction in the levels of Fe-S cluster proteins, including aconitase, NDUFS3 of Complex I, SDHB of Complex II, and Rieske protein of Complex III. On the other hand, no change was observed in proteins without Fe-S clusters [89]. Moreover, during late-stage cardiomyopathy, a further Lonp1 induction and Fe-S protein levels reduction in the heart were observed, suggesting that Fe-S proteins are substrates of Lonp1 activity [89].

Sepuri et al. suggested that Lonp1 is associated with cardiac stress response through the degradation of subunits IVi1 and Vb of Complex IV-cytochrome c oxidase [88]. Previous works reported that, in hypoxic cells and during I/R injury in rabbit hearts, subunits IVi1 and Vb were hyperphosphorylated and downregulated, leading to increased ROS production and myocardium cell death [90,91]. Experiments with recombinant proteins revealed that Lonp1, which is upregulated under ischemia, could effectively degrade phosphorylated IVi1 and Vb subunits. This effect could not be observed with site-specific mutant subunits that could not be phosphorylated. This study postulated that the inhibition of Lonp1 could reduce hypoxic and ischemic damage. Conversely, other studies have indicated that Lonp1-mediated degradation of phosphorylated IVi1 and Vb subunits may enable cardiomyocyte remodeling and reprogramming, hence providing a protective response to ischemic conditions [24].

The role of Lonp1 as protective versus I/R injury has been further strengthened by observations made on a mouse model overexpressing Lonp1. During the reperfusion phase of I/R injury, a burst of ROS occurs, causing oxidative damage to nucleic acids, proteins, and lipids and resulting in the apoptosis and/or necrosis of the myocardium [92,93]. Overexpression of Lonp1 in transgenic mice hearts reduced protein carbonylation and lipid peroxidation, known cardiac I/R injury hallmarks in vivo [94] during ischemia and early reperfusion [61]. Moreover, the activity of aconitase was significantly increased during ischemia and early reperfusion with no change in protein levels, in comparison to wild type mice heart at any I/R time points [61]. These results are in contrast to what seen by Bulteau et al. using a rat model of ischemia and reperfusion, where a transient decline in aconitase activity was observed during early reperfusion [94]. 

Cantu et al. reported that Lonp1 was capable of degrading oxidized aconitase, thus preventing its toxic accumulation and the consequent release of iron, which could lead to the formation of ROS [95]. Thus, transgenic mice overexpressing Lonp1 in the heart were developed to further investigate the role of Lonp1 in maintaining a redox balance in the mitochondrial matrix [61]. Increased aconitase activity in these Lonp1 overexpressing mice during I/R could potentially counteract the augmented formation of hydroxyl radicals during reperfusion.

Lonp1 plays a vital role in controlling heart metabolic adaptability under stressful conditions, enabling cardiomyocytes to transition between alternative energetic pathways [96]. In mature cardiomyocytes, mitochondrial oxidative metabolism provides more >90% of cellular ATP and FAO is the major source of mitochondrial oxidative metabolism, whereas only a small proportion of ATP (around 5%) is synthesized by glycolysis [97]. During cardiac stress and damage, a transition from FAO to glycolysis occurs [98,99,100]. Glycolysis yields less ATP than FAO, and sole reliance on glycolysis is detrimental and increases the risk of cardiac failure. Therefore, strict coordination between glycolysis and FAO is essential. In the heart, Lonp1 minimizes cardiac stress and injury by reprogramming energy metabolism through the regulation of PDH activity and OXPHOS complexes [19,24,48]. PDH forms a complex associated with PDH kinases (PDK 1 to 4), which inhibit PDH by phosphorylating E1α subunit, and PDH phosphatases (PDP 1 and 2), which reactivate PDH by dephosphorylating E1α. Studies in isolated mouse hearts demonstrate that Lonp1 regulates PDH activity by degrading PDK4 dissociated from the PDH complex, with a decreased E1α phosphorylation and subsequent maintenance of PDH in the active state [19]. PDK4 degradation was shown to be dependent on substrate availability [19]. Diet strongly influences this regulation pathway: mice undergoing a high-fat diet, even if only for a single day, showed a PDK4 upregulation, which blunts PDH activity. When mice are returned to a control diet, PDK4 is degraded by Lonp1 [19]. 

## 4. The Functions of Lonp1 in the Skeletal Muscle

Skeletal muscle is one of the largest metabolically active organs and depends critically on mitochondrial quality [101]. As mitochondrial dysfunction is a hallmark of aging and causes several diseases that can affect this tissue, it is essential to maintain the integrity and functionality of the mitochondrial network in muscle fibers. Accumulating evidence, from both in vitro experiments and in vivo models, has demonstrated that Lonp1 is essential for the development of skeletal muscle and the differentiation of myocytes [102]. 

The first lines of evidence of the essential role of Lonp1 in muscle development and functions comes from studies in vivo on *Drosophila melanogaster*, a valid model to study muscle development and age-related deterioration. Defects in locomotion and flight become progressively more evident during the short life of this organism [103,104,105]. Lonp1 knock-out in *Drosophila* resulted in a shorter lifespan and locomotor defects from the earliest moments of life, partially due to an altered respiratory chain function and accumulation of unfolded mitochondrial proteins. These alterations were strictly associated with reduced OXPHOS capacity and decreased ATP production [106,107,108]. Moreover, Lonp1 knockdown leads to a deficit in respiratory chain subunits encoded by the mitochondrial genome [109]. Finally, Lonp1 depletion causes an accumulation of oxidized proteins in the mitochondria of skeletal muscle and can trigger UPRmt response, in agreement with its role as a mitochondrial sensor of cellular stress. Interestingly, these defects appear partially rescued, but not reversed, by the overexpression of ClpP [110].

Studies conducted in vivo on mice reported that Lonp1 loss of function leads to the alterations of mitochondrial ultrastructure and organelle functions [66]. Even heterozygous mice showed a reduction in the levels of Lonp1 in skeletal muscles. Unpublished observations from our group suggest that Lonp1 levels undergo a progressive, age-dependent decline at a faster pace in Lonp1^+/−^ mice than wild type animals and that the sole inactivation of proteolytic activity by introducing a site-specific mutation in the proteolytic dyad of the enzyme causes the same phenomenon. 

In humans, evidence of the crucial role of Lonp1 in skeletal muscle comes from observations on patients with CODAS and CDH. As previously mentioned, Lonp1 mutations cause CODAS syndrome, which invariably presents with hypotonia and mild to moderate motor delay at the muscular level. Some patients also show ptosis and spasticity [46,111], which are other possible signs of muscular defects. Importantly, these symptoms are shared with inherited disorders characterized by mitochondrial defects [16]. Accordingly, muscle biopsies in these patients revealed the presence of inclusions and damaged proteins associated with OXPHOS defects in mitochondria [112]. 

A series of recent studies have elucidated the mechanisms that link the Lonp1 expression of myocyte development and homeostasis. In particular, accumulating evidence indicates that Lonp1 plays a significant role in the regulation of myogenesis by modulating the fine interconnections between autophagy/mitophagy and mitochondrial network and dynamics.

Autophagy is normally stimulated by the onset of myoblast differentiation in order to eliminate damaged organelles and proteins and to maintain correct structure and function of skeletal muscle [113,114]. However, when excessively activated, autophagy provokes the removal of parts of the organelles, proteins, and cytoplasm, exacerbating muscle wasting [115,116,117]. Huang et al. studied immortalized mouse myoblasts and demonstrated that autophagy is blocked as a consequence of Lonp1 downregulation only at the late stage of myoblast differentiation, whereas there were no significant effects of knockdown at the early stage of differentiation [118]. 

During myogenesis, a dynamic and finely regulated process required for both myoblast differentiation and myotube maturation, mitochondria undergo continuous remodeling, mediated by fusion and fission [119]. Mitofusion, mediated by mitofusin 1/2 (Mfn1/2) and optic atrophy-1 (OPA1), and mitofission, mediated by dynamin-related protein 1 (Drp1), are strictly coordinated processes that regulate the mitochondrial network and dynamics [120]. Specifically, Mfn2 increases during myoblast differentiation while Drp1 significantly decreases in mature myotubes [120]. By activating mitofission and inactivating mitofusion at the early stage of myoblast differentiation, undifferentiated cells exhibited a greatly increased number of spherical mitochondria. Conversely, mitofission is inactivated and mitofusion is activated at the later stage of differentiation, creating a highly dense mitochondrial network without the possibility of properly distinguishing single mitochondria [120]. All the changes described are necessary to cope with the increased amount of energy required for myoblast contraction after differentiation [121]. Recent studies report that Lonp1 can modulate mitofusion and fission by activating PINK1/Parkin-dependent mitophagy [122], a crucial process that regulates mitochondrial morphological remodeling during myogenesis by removing damaged mitochondria [123,124]. Conversely, Lonp1 knockdown suppresses the PINK1/Parkin pathway and causes the downregulation of Mfn2 and Drp1 [125,126,127], leading to the accumulation of damaged mitochondria [128,129].

In agreement with these observations, several recent studies show an increase in the levels of PARK7 protein, one of the main Lonp1 substrates, when Lonp1 is specifically knocked down in muscle mitochondria, highlighting a central role of this protein in the regulation of autophagy upon the loss of Lonp1 [130,131].

Taken together, these observations underline the importance of autophagic/mitophagic process activation to counteract the aberrant accumulation of mitochondrial damaged proteins induced by Lonp1 downregulation, favouring skeletal muscle clearance and preserving muscle mitochondrial functions [101].

In both humans and mice, Lonp1-dependent autophagy pathway is associated with skeletal muscle loss and weakness induced by disuse. This process is triggered by decreased activity or aging, and muscles are mainly characterized by reduced fiber size and strength [101]. This condition represents one of the major health problems because it can worsen the quality of life and increase mortality. Variations in physical activity and an early ablation of Lonp1, which does not affect the protein levels of Mfn1 and Opa1, can affect skeletal muscle mass by altering the delicate balance between protein synthesis and protein degradation in mitochondria [132,133,134]. In a study by Xu and colleagues, Lonp1 loss-of-function effects on muscle were studied both in vivo by generating a mouse with the skeletal muscle-specific deletion of Lonp1 and in vitro where skeletal myocytes in culture were forced to differentiate into myotubes. The results revealed that the effects of Lonp1 ablation in myotubes recapitulated the effect in vivo and were characterized by the reduction of fully assembled respiratory complexes IV and mitochondrial respiration alterations. Conversely, they did not detect modifications in the ratio between ubiquitin-mediated protein degradation and protein synthesis [101]. These findings indicate that Lonp1 is responsible for skeletal muscle maintenance during muscle disuse. Therefore, a correlation exists between safeguarding the mitochondrial network and protecting skeletal muscle mass and strength [101]. Genetic ablation of Lonp1 specifically in skeletal muscle originates a precocious aging phenotype, in agreement with its role in maintaining mitochondrial homeostasis.

Lonp1 seems to be essential also in the maintenance of energy homeostasis following metabolic stresses in mammals. Several studies reported that Lonp1 ablation in mouse muscle induces UPRmt activation, leading to adipose tissue and liver metabolism modulation. Functionally, in this model not only the well-known mitochondrial protein turnover alterations can be observed, but also improved insulin resistance, reduced liver steatosis and prevention of high-fat diet (HFD)-induced obesity [135]. Furthermore, following Lonp1 ablation, muscles are forced to secrete myokines, especially FGF21 and GDF15, to exert endocrine effects, which can contrast diet-induced obesity and generates a positive effect on metabolism [136,137,138,139,140]. 

Guo and colleagues showed that, upon HFD feeding, Lonp1 knockout mice are characterized by reduced lean and fat mass, leading to lower body weight [101,135], smaller adipocytes in adipose tissue, higher mitochondrial respiration and oxidation, increased expressions of genes related to thermogenesis, and lipolysis. Conversely, there is a decrease in fatty acid synthase (FASN) protein levels and de novo lipogenesis, leading to protection from diet-induced hepatic steatosis. Moreover, food intake is increased indicating a deficit in conversion of food into body mass when Lonp1 is downregulated. Therefore, Guo and colleagues proposed an important role of Lonp1 in modulating muscle adaptation under excess nutrient stress [135]. In addition, Lonp1 skeletal muscle-specific ablation upon HFD feeding triggers the induction of several UPRmt markers, suggesting that muscle UPRmt signals may have a protective effect, as they can regulate the communication between adipose tissue and the liver and alleviate dietary obesity. Previous studies hypothesized a possible UPRmt regulation through Atf4 in mammals [141,142,143,144,145,146]. Additionally, Guo et al. generated muscle-specific Lonp1/ATF4 KO mice to investigate if ATF4 ablation would be sufficient to eliminate the metabolic and genomic reprogramming prompted by the loss of Lonp1 [135]. The authors observed the reduced expression of amino acid metabolism genes and relevant increase of one-carbon metabolism genes, suggesting the importance of ATF4 in the regulation of metabolism and response to stimuli in skeletal muscle. However, ATF4 knocked down did not affect the expression of the main genes regulated when Lonp1 is downregulated; in addition, liver and adipose metabolic phenotypes were not significantly different between the LONP1 mKO and Lonp1/ATF4 KO mice model under HFD conditions. Thus, these results provide substantial evidence that UPRmt mainly acts through ATF4-independent mechanisms, and the Lonp1 effects on genomic reprogramming are independent of ATF4 in skeletal muscle [135]. Figure 1 summarizes the main functions and the essential role played by Lonp1 in cardiomyocytes and myocytes. 

## 5. Lonp1 and Aging: A Role in Sarcopenia

Lon protease has a significant influence on the aging process [147,148,149]. Reports have shown that deleting *Pim1* in *S. cerevisiae* leads to faster aging [150] and *C. elegans* lonp-1 mutants had significantly shorter mean lifespan than wild type worms [151]. On the other hand, its overexpression in the filamentous ascomycete *Podospora anserina* extends health-span and lifespan [152]. The increase of *P. anserina* Lonp1 levels determined an increased lifespan without compromising fitness, reduced carbonylated proteins, and better resistance to external oxidative stress such as H_2_O_2_-induced stress [152]. In contradiction to the findings in *P. anserina*, research with *D. melanogaster* demonstrated that decreasing or increasing the levels of Lonp1 shortens the lifespan, suggesting that Lonp1 needs to be present in normal amounts for optimal longevity in this species [109,153]. Extending the lifespan in *Drosophila* appears to be associated with the capability of Lonp1 to confer resistance to oxidative stress [154], which aligns to the fact that the detrimental effects of ROS are responsible for the mitochondrial impairment observed during oxidative stress in aged cell [23]. Interestingly, resistance to ROS in *Drosophila* is sex-specific: female-specific resistance to H_2_O_2_ stress and male-specific resistance to superoxide stress require Lonp1 activity while an overexpression or silencing of Lonp1 in the same model was found detrimental to lifespan [154].

Results obtained on rat cardiac muscle showed that an upregulation of Lonp1 occurs during aging. Moreover, although Lonp1 expression also increased in aged mouse hearts, its protease activity remained stable in heart mitochondria during aging [23]. These findings suggest that Lonp1 efficiency may decrease with aging due to persistent oxidative stress stimuli. Thus, Lonp1 undergoes an age-related functional decline, leading to the accumulation of oxidatively inactivated proteins, similar to what is observed in heart failure.

As stated before, aging triggers several alterations related to the downregulation of OXPHOS complex proteins, the major marker of mitochondrial biogenesis, and UPRmt response, such as the reduced gene expression of ATP-dependent proteases and the ATF4 [102]. 

Mitochondrial dysfunction contributes to mitochondrial aging in myocytes, and mitophagy plays a central role in delaying age-related disorders. Consequently, mitophagy-deficient cells exhibit a greater number of damaged mitochondria and a significant increase in ROS production, a phenomenon that underlines the tight association between mitophagy, aging, and oxidative stress [129,155].

Aging commonly induces the reduction of both Lonp1 protein levels and proteolytic activity in mouse skeletal muscle because the progressive degeneration of mitochondrial capacity causes the reduction of mtRNAs expression and mtDNA levels. In addition, there is a decrease in the protein levels of OXPHOS components and an accumulation of oxidatively modified proteins [21,156,157,158], leading to higher ROS production and oxidative stress [159].

In this regard, Guo and colleagues analyzed skeletal muscle in an in vivo aged mice model and reported an upregulation of mitochondrial-related genes expression when UPRmt genes are preserved, indicating that UPRmt is linked to mitochondrial metabolism in aged mice. Therefore, control and maintenance of mitochondrial functions are essential to promote proper activation of UPRmt in the skeletal muscle of aged mice [102]. In this scenario, mitonuclear imbalance can be prompted by physical activity, leading to an overexpression of several UPRmt markers, including Lonp1. Regarding Lonp1-deficient myocytes, they exhibit a pathological phenotype characterized by miniature mitochondria and giant mitochondria with electron-dense inclusion bodies and large empty vacuoles [8], which are autophagocytosed with greater difficulty because of their size [8,160]. Consequently, giant mitochondria gradually accumulate in senescent myocytes, triggering several adverse events, such as altered mitochondrial proteins [161,162] or homoplasmic mtDNA point mutations or deletions [163,164], probably resulting in decreased inner membrane potential and a drop in energy production [165]. The phenotype related to Lonp1 depletion described above is the same as that normally found in aging tissue, highlighting the involvement of Lonp1 downregulation in the aging process. Interestingly, Koltai and colleagues suggested that moderate physical activity can improve mitochondrial quality control and biogenesis, minimizing some skeletal muscle deficits due to aging. Thus, moderate intensity exercise could counteract the physiological age-induced decline of Lonp1 [166,167]. The main effects observed on heart and skeletal muscle of Lonp1 decline observed with aging are summarized in Figure 2.

In conclusion, the aging process can be induced, prompted, or accelerated when the delicate equilibrium among mitochondrial damage, mitophagy, mitochondrial respiration, and ROS production is unbalanced, either in physiological or pathological conditions.

The main findings concerning the role of Lonp1 in heart and skeletal muscle obtained by the experimental models described in the Section 3, Section 4 and Section 5 are summarized in Table 2.

## 6. Conclusions and Future Perspective

The discovered and described functions of Lonp1 have undergone a great expansion in the last 15 years and indicate an increasingly important role of this protein in the regulation of mitochondrial homeostasis and of the whole cell. This is especially true in the case of energy-demanding tissues, such as myocardium or skeletal muscle tissue. The picture that emerges from the observations and studies discussed in this review is that Lonp1 is an essential protein in these tissues not for functions specifically related to cardiomyocyte or myocyte activity, but rather because some Lonp1 activities, such as its role as a quality control for mitochondrial proteins, are particularly important in these tissues. In a similar way, Lonp1 is not a driver of aging per se in these tissues but can indirectly affect the aging of heart and skeletal muscle by modulating several quality control systems of the mitochondria, whose proper activity are particularly important for the maintenance of organelle- and cell-function. As new emerging activities of Lonp1 have been described in recent years (for instance, its interaction with mitochondrial sirtuins [168], or its relevance in steroid synthesis pathways in the mitochondria [169]) it would be interesting in the future to determine if changes to Lonp1 activities can have an impact on the skeletal muscle of heart homeostasis and function.

## Figures and Tables

**Figure 1 antioxidants-12-00598-f001:**
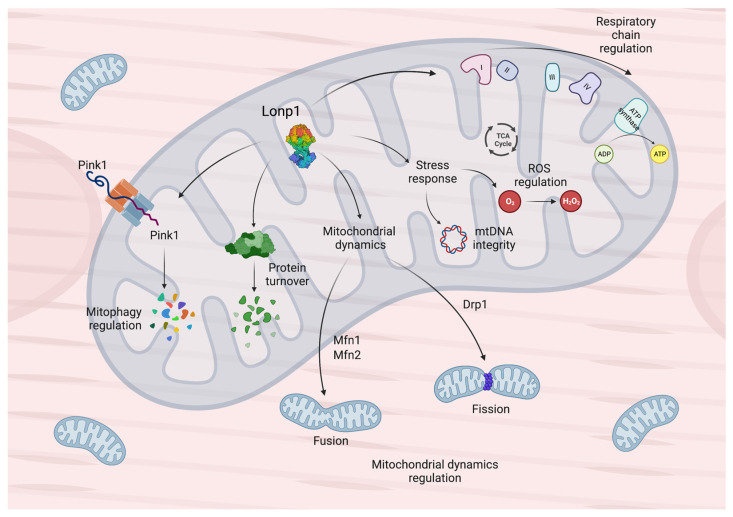
Mitochondrial functions of Lonp1 in myocytes and cardiomyocytes. In mitochondria, Lonp1 (i) regulates mitophagy through Pink1 degradation; (ii) regulates protein turnover by degrading misfolded or damage proteins; (iii) modulates mitochondrial dynamics through the action of Mfn1 and Mfn2, which promote fusion, and Drp1, which promotes fission; (iv) regulates stress response by preserving mtDNA integrity and modulating ROS production; (v) modulates respiratory chain complexes activity.

**Figure 2 antioxidants-12-00598-f002:**
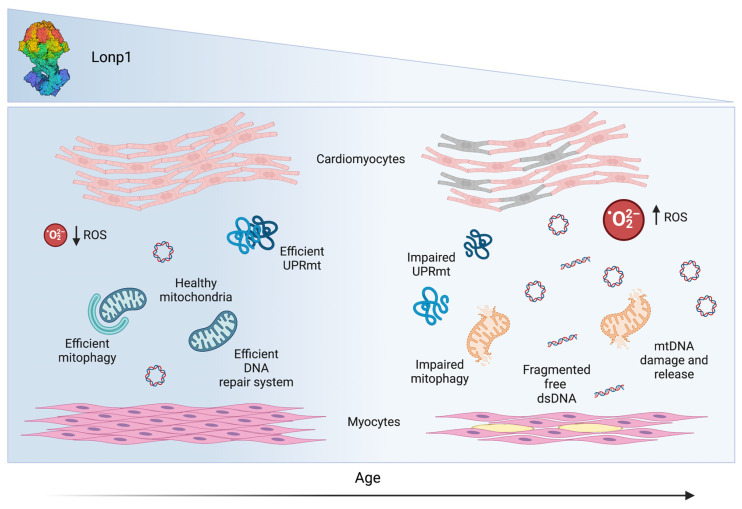
Age-related Lonp1 effects on cardiomyocytes and myocytes. In the mitochondria of cardiomyocytes and myocytes, Lonp1 plays an important role in counteracting ROS production and maintaining mitochondrial homeostasis through efficient mitophagy, DNA repair system, and UPRmt response. During aging, Lonp1 levels decrease, leading to higher ROS levels, impaired mitophagy, release of damaged mtDNA and fragmented dsDNA, and impaired UPRmt response, a series of effects that can favour myocyte damage in skeletal muscle and sarcopenia.

**Table 2 antioxidants-12-00598-t002:** Effects of experimental modulation of Lonp1 expression on heart and skeletal muscle.

Organ	Experimental Model	Lonp1 Levels	Effects	References
Heart	Mouse cardiomyocytes	Normal expression	Glucose, FAO enzymes and PDH levels modulation in maturing cardiomyocytes	[48]
	Rat cardiomyocyte H9c2 cells	Overexpression	Apoptosis under normoxic conditions	[74,75]
		Downregulation	Mitigation of cell death induced by hypoxia	[74,75]
	Mouse	Knock out	Severe defective heart development; embryonic lethality; Reduction nof cardioprotective effect of IPC; Increment in myocardial infarct size	[66,67,74,75,84]
		Downregulation	fragmentation of mitochondria; cardiomyocytes aberrant metabolic reprogramming; cardiomyopathy; HF	[67]
		Normal expression	Reduction of cardiac stress and injury by reprogramming energy metabolism, thorough the regulation of PDH activity and OXPHOS complexes	[19,24,48,83]
		Upregulation	Reduction of oxidative damage; preservation of redox state of mitochondria; reprograming of mitochondrial bioenergetics; reduction of complex I activity, ROS production, and cardiac cell death	[61]
		Overexpression	Reduction of protein carbonylation and lipid peroxidation during ischemia and early reperfusion	[61,94]
Skeletal muscle	Drosophila	Knockout	Locomotion defects; alteration of respiratory chain function; accumulation of unfolded and oxidized mitochondrial proteins; reduction of OXPHOS capacity and ATP production; stimulation of UPRmt response	[106,107,108,109]
	Mouse C2C12 cells	Knockout	Suppression of PINK1/Parkin pathway; alterations of mitochondrial dynamics; accumulation of damaged mitochondria	[128,129]
	Immortalized mouse myoblasts	Downregulation	Block of autophagy, only at the late stage of myoblast differentiation	[118]
	Mouse myotubes	Knock out	Reduction of fully assembled respiratory complexes IV; alterations of mitochondrial respiration	[101]
	Mouse	Knock out	Alteration of mitochondrial ultrastructure and organelle functions; hypotonia, mild to moderate motor delay; Release of myokines; Reduction of lean and fat mass and lower body weight	[16,25,66,101]
	Mouse under high-fat diet (HFD)	Knock out	Activation of UPRmt; alterations of mitochondrial protein turnover; improvement of insulin resistance; reduction of liver steatosis; prevention of high-fat diet HFD–induced obesity	[135]

## Data Availability

Not applicable.

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
