# Peer review of "The Role of Lonp1 on Mitochondrial Functions during Cardiovascular and Muscular Diseases"

_antioxidants, 2023, doi:10.3390/antiox12030598_

Round 1

Reviewer 1 Report

The review written by Zanini and Selleri describes the role of Lonp1 in mitochondrial function during health and disease.

Generally, the review is well-written and clear to the reader.

I suggest the authors to take into consideration the following points:

Title

Please justify the use of the term cardiorespiratory in the main title as it is not mentioned elsewhere.

Abstract

·        Please revise the English text of the abstract.

e.g. FROM “The mitochondrial protease Lonp1 is a multifunctional enzyme that regulates crucial 14 mitochondrial functions, including degradation of oxidized proteins, the folding of imported 15 proteins and the maintenance the correct number of copies of mitochondrial DNA. TO The mitochondrial protease Lonp1 is a multifunctional enzyme that regulates crucial 14 mitochondrial functions, including degradation of oxidized proteins, folding of imported 15 proteins and maintenance of the correct number of copies of mitochondrial DNA

e.g. FROM “During heart development, Lonp1 allows the metabolic shift from anaerobic glycolysis 18 to mitochondrial oxidative phosphorylation and knock out of Lonp1 arrest heart development and 19 determines cardiomyocyte apoptosis.” TO During heart development, Lonp1 allows the metabolic shift from anaerobic glycolysis 18 to mitochondrial oxidative phosphorylation.  Knock out of Lonp1 arrests heart development and 19 determines cardiomyocyte apoptosis

·        Please use the present tense overall your abstract.

Introduction

·        Lines 55-59: “In this review, we explore the primary physiological functions of Lonp1 in human 55 cells, its role in the heart and skeletal muscle as a stress response protein, emphasizing its 56 involvement in the oxidative stress reaction, and the functional outcomes of the age- 57 related Lonp1 expression decrease in the onset of cardiovascular diseases and skeletal 58 muscle functional decline

To my understanding, authors present primarily the physiological and pathophysiological functions of Lonp1. Secondly, they describe its role in heart and skeletal muscle health and disease, finally they discuss the role of Lonp1 in the aging process. Considering that I would slightly rephrase this paragraph to make it clearer to the reader.

·        Please add references to lines

39

40

41

54

Section 2: General features and functions of Lonp1

·        In this section, the authors describe not only the general function of Lonp1 but also its implications in diseases/imbalanced conditions. Please rephrase the title of the section

·        I suggest the authors to summarize their literature search on Lonp1 function as a Table

Section 5: Lonp1 and aging: a role in sarcopenia?

·        Please rephrase the tile of the section without the “question mark”

Section 5: Conclusions and future perspective

·        This corresponds to section 6

·        Is the title of Table 1 correct? (Table 1.Role of mtDNA as inflammatory molecules in human pathologies described in this re- 569 view)

·        Please move Table 1 to another section of the paper and rephrase the conclusion section (Conclusion section should summarize the main messages of the manuscript and include only “conclusive” remarks)

OTHER GENERAL COMMENTS

·        Please write in vitro and in vitro in italics

·        Figure 1 and Figure 2 are not mentioned in the main text. Please include that and insert the Figures only after having mentioned them in the text.

Author Response

The review written by Zanini and Selleri describes the role of Lonp1 in mitochondrial function during health and disease.

Generally, the review is well-written and clear to the reader.

ANSWER: We thank the reviewer for appreciating our work. 

I suggest the authors to take into consideration the following points:

Title

Please justify the use of the term cardiorespiratory in the main title as it is not mentioned elsewhere.

ANSWER: The title was initially agreed upon with the Journal editor, but the critical review of the scientific literature did not highlight studies related to respiratory diseases involving Lonp1. We have slightly changed the title to make it more faithful to the content of the text as follows: “The role of Lonp1 on mitochondrial functions during cardiovascular and muscular diseases”.

Abstract

Please revise the English text of the abstract.

e.g. FROM “The mitochondrial protease Lonp1 is a multifunctional enzyme that regulates crucial 14 mitochondrial functions, including degradation of oxidized proteins, the folding of imported 15 proteins and the maintenance the correct number of copies of mitochondrial DNA. “ TO The mitochondrial protease Lonp1 is a multifunctional enzyme that regulates crucial 14 mitochondrial functions, including degradation of oxidized proteins, folding of imported 15 proteins and maintenance of the correct number of copies of mitochondrial DNA

e.g. FROM “During heart development, Lonp1 allows the metabolic shift from anaerobic glycolysis 18 to mitochondrial oxidative phosphorylation and knock out of Lonp1 arrest heart development and 19 determines cardiomyocyte apoptosis.” TO During heart development, Lonp1 allows the metabolic shift from anaerobic glycolysis 18 to mitochondrial oxidative phosphorylation.  Knock out of Lonp1 arrests heart development and 19 determines cardiomyocyte apoptosis

ANSWER: We rephrased the abstract as requested by the reviewer: “The mitochondrial protease Lonp1 is a multifunctional enzyme that regulates crucial mitochondrial functions, including degradation of oxidized proteins, folding of imported proteins, and maintenance of the correct number of copies of mitochondrial DNA. A series of recent studies has put Lonp1 at the center of the stage in the homeostasis of cardiomyocytes and muscle skeletal cells. During heart development, Lonp1 allows the metabolic shift from anaerobic glycolysis to mitochondrial oxidative phosphorylation.  Knockout of Lonp1 arrests heart development and determines cardiomyocyte apoptosis. In adults, Lonp1 acts as a cardioprotective protein, as its upregulation mitigates cardiac injury by preventing oxidative damage of proteins and lipids, and by preserving mitochondrial redox balance. In skeletal muscle, Lonp1 is crucial for cell development, as it mediates the activation of PINK1/Parkin pathway needed for proper myoblast differentiation. Skeletal muscle-specific ablation of Lonp1 in mice causes reduced muscle fiber size and strength, due to the accumulation of mitochondrial-retained protein in muscle. Lonp1 expression and activity decline with age in different tissues, including skeletal muscle, and it is associated with a functional decline and structural impairment of muscle fibers. Aerobic exercise increases Unfolded protein response markers -including Lonp1 - in the skeletal muscle of aged animals and are associated with muscle functional recovery. Finally, mutations of Lonp1 cause a syndrome named CODAS (Cerebral, Ocular, Dental, Auricular and Skeletal anomalies) characterized by impaired development of multiple organs and tissues, including myocytes. CODAS patients show hypotonia and ptosis, indicative of skeletal muscle reduced performance. Overall, this body of observations points Lonp1 as a crucial regulator of mitochondrial functions in the heart and in skeletal muscle.”

Please use the present tense overall your abstract.

ANSWER: As requested, we changed past tense to present tense when needed.

Introduction

Lines 55-59: “In this review, we explore the primary physiological functions of Lonp1 in human 55 cells, its role in the heart and skeletal muscle as a stress response protein, emphasizing its 56 involvement in the oxidative stress reaction, and the functional outcomes of the age- 57 related Lonp1 expression decrease in the onset of cardiovascular diseases and skeletal 58 muscle functional decline

To my understanding, authors present primarily the physiological and pathophysiological functions of Lonp1. Secondly, they describe its role in heart and skeletal muscle health and disease, finally they discuss the role of Lonp1 in the aging process. Considering that I would slightly rephrase this paragraph to make it clearer to the reader.Please add references to lines

39

40

41

54

ANSWER: New references have been provided, as requested, and the paragraph has been slightly rephrased, as follows:

“In this review, we explore primarily the physiological and pathophysiological functions of Lonp1 in humans. Then, we discuss its role in heart and skeletal muscle in health and disease, emphasizing its involvement in the response to oxidative stress. Finally, we discuss the role of Lonp1 in the aging process, and the functional outcomes of the age-related Lonp1 expression decrease in the onset of cardiovascular diseases and skeletal muscle functional decline.”

 These are the new references added:

  1. Galluzzi, L.; Kepp, O.; Trojel-Hansen, C.; Kroemer, G. Mitochondrial control of cellular life, stress, and death. Circ Res 2012, 111, 1198-1207, doi:10.1161/CIRCRESAHA.112.268946.
  2. Delbridge, L.M.D.; Mellor, K.M.; Taylor, D.J.; Gottlieb, R.A. Myocardial stress and autophagy: mechanisms and potential therapies. Nat Rev Cardiol 2017, 14, 412-425, doi:10.1038/nrcardio.2017.35.
  3. Giorgi, C.; Marchi, S.; Pinton, P. The machineries, regulation and cellular functions of mitochondrial calcium. Nat Rev Mol Cell Biol 2018, 19, 713-730, doi:10.1038/s41580-018-0052-8.
  4. Bonora, M.; Wieckowski, M.R.; Sinclair, D.A.; Kroemer, G.; Pinton, P.; Galluzzi, L. Targeting mitochondria for cardiovascular disorders: therapeutic potential and obstacles. Nat Rev Cardiol 2019, 16, 33-55, doi:10.1038/s41569-018-0074-0.

6 .        Lopez-Crisosto, C.; Pennanen, C.; Vasquez-Trincado, C.; Morales, P.E.; Bravo-Sagua, R.; Quest, A.F.G.; Chiong, M.; Lavandero, S. Sarcoplasmic reticulum-mitochondria communication in cardiovascular pathophysiology. Nat Rev Cardiol 2017, 14, 342-360, doi:10.1038/nrcardio.2017.23.

  1. Bravo-San Pedro, J.M.; Kroemer, G.; Galluzzi, L. Autophagy and Mitophagy in Cardiovascular Disease. Circ Res 2017, 120, 1812-1824, doi:10.1161/CIRCRESAHA.117.311082.

Section 2: General features and functions of Lonp1

In this section, the authors describe not only the general function of Lonp1 but also its implications in diseases/imbalanced conditions. Please rephrase the title of the section

ANSWER: As requested, we have changed the title of the section from 'General features and functions of Lonp1' to 'General features and functions of Lonp1 and its implication in diseases'

I suggest the authors to summarize their literature search on Lonp1 function as a Table

ANSWER: We added a new Table (Table 1) with the main functions of Lonp1 reported in the First section of the manuscript

Section 5: Lonp1 and aging: a role in sarcopenia?

Please rephrase the tile of the section without the “question mark”

ANSWER: We removed the question mark, as requested.

Section 5: Conclusions and future perspective

This corresponds to section 6

ANSWER: The reviewer is right, we corrected it

Is the title of Table 1 correct? (Table 1. Role of mtDNA as inflammatory molecules in human pathologies described in this re- 569 view)

ANSWER: We apologized for the mistake, that we promptly corrected. The right title is “Effects of experimental modulation of Lonp1 expression on heart and skeletal muscle”

Please move Table 1 to another section of the paper and rephrase the conclusion section (Conclusion section should summarize the main messages of the manuscript and include only “conclusive” remarks)

 ANSWER: Table 1 has been moved to Section 5

OTHER GENERAL COMMENTS

 Please write in vitro and in vitro in italics

ANSWER: To the best of my knowledge, the policy of MDPI is to not write in vitro and in vitro in Italics, and so we followed their guidelines. Should the journal editor tell us to change it to Italics, we will be pleased to do that.

Figure 1 and Figure 2 are not mentioned in the main text. Please include that and insert the Figures only after having mentioned them in the text.

ANSWER: We mentioned the Figures in the paper, and moved them immediately after being cited.

Reviewer 2 Report

The manuscript entitled “The role of Lonp1 on mitochondrial functions during cardio-respiratory and muscular diseases” by Zanini et al summarizes the role of Lonp1 on mitochondrial functions in important disease. The manuscript is well written and organized. However, a lot of typo and grammatical mistakes should be corrected. For example formatting in the title itself.

The manuscript can be accepted after minor revisions.

Author Response

We thank the reviewer for her/his comments. As requested, we carefully checked and amended the manuscript for typos and grammar mistakes, with the assistance of an English native speaker.

Reviewer 3 Report

Zanini et al. have written a very thorough review of the many mechanisms of action of the Lon protease 1 on mitochondrial function in cardiac and skeletal muscle.  This is an excellent, informative, and well-written review that is not duplicated in the literature.  The review has excellent diagrams and a table that help the reader further by displaying the information graphically.

Author Response

We thank the reviewer for appreciating our work.